# OpenReview forum: "LoFiT: Localized Fine-tuning on LLM Representations"
_NeurIPS.cc/2024/Conference — NeurIPS 2024 poster_

### Official Review · Reviewer_Yw79 · 2024-07-05

**Soundness:** 2
**Presentation:** 4
**Contribution:** 2
**Rating:** 6
**Confidence:** 4

**Summary:**

The paper proposes LoFiT, a procedure for *localized* fine-tuning of LLMs. LoFiT chooses a task-specific subset of attention heads by tuning head-wise learnable scales and selecting the heads with largest scales (by absolute value). After that, the algorithm tunes the biases for the chosen attention heads to solve the chosen task. The paper includes experiments on three benchmarks using LLMs from Llama-2 and Gemma families. Authors compare LoFiT against alternative head selection methods and against several PEFT algorithms (LoRA and RED). The paper also analyzes the impact of heads found by LoFiT in different scenarios.

**Strengths:**

1. Authors propose a very simple algorithm for selecting heads that seemingly works well (within the 3 chosen tasks). The fact that it is simple is a significant advantage: such algorithm would be easy to modify or reuse in other circumstances.

2. The paper includes several ablations and sub-analyses that answer the questions that arise when reading it. This is a sign that the experiments are well structured.

3. The paper is generally well written, well organized and easy to read.

**Weaknesses:**

My main concern about the paper is that the main evaluations are limited to 3 tasks (TruthfulQA, MQuAKE, CLUTRR). This makes it unclear if LoFiT is generally applicable in place of PEFT methods or if it is only competitive for a specific type of fine-tuning task. If latter is the case, the paper would be substantially stronger if it explained which tasks LoFiT is capable of and where it isn't. If latter is the case, it would be best to include more tasks from among PEFT papers and general LLM fine-tuning scenarios. For instance, the papers you compare against also evaluate on GSM8K, MMLU,  MNLI-m, RTE, (super)GLUE for smaller models and more.

Naturally, I do not suggest that you evaluate on *all* of the benchmarks, but the paper could be strengthened by either showing that LoFiT generalizes to more tasks or describing (and demonstrating) the types of tasks where it works poorly. This would help readers understand where to use LoFiT (or its components) instead of other intervention / peft methods.

**Questions:**

**Q1:** To the best of my understanding, the LoFiT algorithm modifies attention heads and keeps FFN / MLP layers intact. In contrast, several popular model editing algorithms (e.g. MEMiT [1] or Transformer-Patcher[2]) focus on updating FFN / MLP layers alone. When should one focus on editing attention layers or FFN layers. Are these interchangable or is there a type of tasks that do best with attention editing?

- [1] https://arxiv.org/abs/2210.07229
- [2] https://arxiv.org/pdf/2301.09785


**Q2:** when selecting the heads, do you need to tune A to convergence to get good scales? If not, how many steps / training tokens are required?


**Q3:** in your experiments how to choose the total number number of heads to be modified? What happens to LoFiT if you choose substantially more of fewer heads?

**Q4:** when reporting the difference in the number of trainable parameters , how do you count the choice of which heads are modified towards the total number of parameters?


> L137 ‘Evaluation is exact match (EM).”

Possibly missing a noun (e.g. evaluation *metric* or *criterion*)

> L294 Finally, we note that the method we present here requires the ability to take gradients, giving it a similar memory footprint as other PEFT methods and making it only usable on open-weight models.

Technically speaking, this statement is false: owners of closed-weight models can use these methods on their models and have, in past, been known to implement some fine-tuning as a service for users (e.g. see openai api finetuning, tunethemodel and others).

**Limitations:**

Authors have sufficiently addressed the limitations of the proposed method. The work can be improved by describing the limitations of the evaluation methodology in more detail.

---

> ### Author Rebuttal · Authors · 2024-08-06
>
> We appreciate your detailed comments and valuable feedback!
>
>
> > My main concern about the paper is that the main evaluations are limited to 3 tasks (TruthfulQA, MQuAKE, CLUTRR). This makes it unclear if LoFiT is generally applicable in place of PEFT methods or if it is only competitive for a specific type of fine-tuning task. [...] It would be best to include more tasks from among PEFT papers and general LLM fine-tuning scenarios.
>
>
> We appreciate your suggestions. Please see the general response for the additional experiments we conducted based on this!
>
>
> > LoFiT algorithm modifies attention heads and keeps FFN / MLP layers intact. In contrast, several popular model editing algorithms focus on updating FFN / MLP layers alone. [...] When should one focus on editing attention layers or FFN layers. Are these interchangable or is there a type of task that does best with attention editing?
>
> In our preliminary experiments, we found that for the tasks in our paper, fine-tuning on attention heads or on MLP layers led to similar performance for most fine-tuning methods. In addition, attention heads have been shown to serve as important model components for truthfulness, QA, and reasoning from the interpretability literature (e.g. Lieberum et al., 2023), so we ended up updating attentions alone. Theoretically, LoFiT can be adapted to be applied to MLP layers in a similar fashion and we can explore this in future versions, but we consider attention heads as a better component for more granular interpretability analyses.
>
> > When selecting the heads, do you need to tune A to convergence to get good scales? How many steps / training tokens are required?
>
>
> Scaling factors (A) need to be trained to convergence. However, we would like to emphasize that the scaling factors are only learned to select important heads and this is much easier than actually learning the task: in our preliminary experiments, we found that the head selection process can converge in fewer epochs and is less sensitive to random seeds and hyperparameters (as shown in Figure 4 of our paper) compared with the final fine-tuning step.
>
>
> > In your experiments how to choose the total number of heads to be modified? What happens to LoFiT if you choose substantially more fewer heads?
>
>
> Please see the results in the general response.
>
>
> > When reporting the difference in the number of trainable parameters, how do you count the choice of which heads are modified towards the total number of parameters?
>
> We only counted the bias offsets because the bias offsets are the only learned parameters that will be used during the inference time after fine-tuning is finished. Our primary consideration is the statistical efficiency of the final learned model, which depends on how many parameters are tuned as opposed to the total number of parameters that need to be touched during (two-stage) training. This is also common in model compression literature (e.g. The prune-retrain paradigm [1]). We note that even counting the number of learned scaling factors, LoFiT only optimizes approximately **half** of the parameters of RED and **3%** of LoRA.
>
>
> > Typos in L137 and L294; The work can be improved by describing the limitations of the evaluation methodology in more detail.
>
>
> Thanks for the precious suggestions! We will revise the typos and include a further discussion of our evaluation methodology in the limitations section of our revision.
>
> References:
>
> [1] Zimmer et al., 2023. PERP: Rethinking the Prune-Retrain Paradigm in the Era of LLMs.

---

> > ### Comment · Reviewer_Yw79 · 2024-08-14
> > **On Author Response**
> >
> > I thank the authors for answering my questions and appreciate the additional evaluations. I have no further questions. With that in mind, I have raised my score by a notch.

---

### Official Review · Reviewer_LqFt · 2024-07-09

**Soundness:** 4
**Presentation:** 4
**Contribution:** 3
**Rating:** 7
**Confidence:** 3

**Summary:**

The paper introduces Localized Fine-Tuning (LoFiT) - a two step method that involves (1) localizing attention heads that are important for a given task, and (2) learning an additive intervention for each important attention head. The authors evaluate the method over various tasks, and show that LoFiT outperforms other inference-time intervention methodologies, and is competitive with PEFT methods despite being much more parameter-efficient.

**Strengths:**

- The paper is well-written.
	- I found the paper easy to read - it is clear and well-organized.
- Strong results (Section 5 & 6)
	- LoFiT outperforms other inference-time intervention methods by a very significant margin (Table 1).
	- LoFiT is competitive with PEFT methods, despite being more parameter-efficient (Table 3).
- Interesting additional investigations (Section 5 & 6)
	- The authors go beyond mere evaluation, and ask interesting follow up questions.
	- They show that localization is important by comparing to a baseline of selecting random heads, that the set of important heads are generally task specific, and that LoFiT shows promise in generalizing out of distribution compared to other methodologies.

**Weaknesses:**

- Could benefit from more thorough comparison with ITI
	- The methodology is very similar to ITI, and as such the difference in performance (Table 1) is surprising. It would be helpful if the authors could explore why LoFiT is so much more effective than ITI.
	- One possible way to explore this would be to investigate the learned bias vectors directly. Are the bias directions found by LoFiT similar to those found using ITI? Are they very different? Do they have similar magnitudes?
- Lacks detail in inference-time intervention baselines
	- Appendix C.2 and D.2 provide some limited details, but it seems important to give more detail, to convince a reader that these baseline methodologies were evaluated properly. Dataset construction is very important for contrastive pair methods, as are the hyperparameters $\alpha$ and the layer $l$ of the intervention (for RepE). Values for $\alpha$ are given, but it would be good to give more detail as to how these values were selected.
- Head-selection baseline
	- I am curious to know if learning $A_{l}^{i} \in \mathbb{R}^{d_{head}}$ is necessary, or whether one could simply learn a scalar $A_{l}^{i} \in \mathbb{R}$ to weight the entire output of head $(l, i)$. This would seem to simplify the method considerably, requiring optimization over only one parameter per head in the first phase.
	- If the simplified version does not perform as well, then I think the method as presented would be better justified.
	- I am also curious to know how the method behaves as $K$ is altered. Is it increasingly effective as $K$ increases? Does the performance difference plateau once $K$ is increased past some value? Concretely, a figure which has $K$ on the x-axis, and performance on the y-axis would be informative.

**Questions:**

- See the weaknesses section.

**Limitations:**

- The authors acknowledge the following limitations:
	- The paper only focuses on English-language evaluation, with short contexts.
	- The paper only explores 3 particular benchmarks. I think this one is particularly salient as a limitation.
	- The paper only evaluates models up to 13B parameters, and results may not extend to larger models.

---

> ### Author Rebuttal · Authors · 2024-08-06
>
> We appreciate your detailed comments and valuable suggestions on our work.
>
> > Could benefit from a more thorough comparison with ITI [...] It would be helpful if the authors could explore why LoFiT is so much more effective than ITI.
>
> We think that the main performance gain of LoFiT over ITI comes from two factors. First, our head selection step selects a better set of heads than the probing method of ITI (see the ITI-heads results in Table 2 of our paper). Second, the end-to-end optimization of our bias tuning method yields more stable intervention vectors than the heuristic-based vector extraction of ITI.
>
> > Are the bias directions found by LoFiT similar to those found using ITI? Are they very different? Do they have similar magnitudes?
>
> We conduct an analysis on the cosine similarity of learned biases between ITI and LoFiT on TruthfulQA with Llama-2-7B. We used the top 48 heads from ITI and learned offset vectors using ITI and LoFiT. Results can be found in the table below. On the same set of heads, LoFiT learns very different offset vectors from ITI.
> |  | Cosine Similarity with LoFiT Biases |  |
> |---|---|---|
> |  | mean | std |
> | ITI: Mass mean shift | 0.0515 | 0.1890 |
> | ITI: Probe weight direction | 0.0427 | 0.1734 |
>
> > Detail of inference-time intervention baselines, including the hyperparameters
>
> We tuned the hyperparameters $\alpha$ for ITI, and the scaling factor $\alpha$ and layer $l$ for RepE on the validation set of each dataset and selected them by the validation accuracy, as suggested by the original papers. We will include the details on the layer $l$ for RepE in our revision of the paper.
>
> > Head-selection baseline: Whether one could simply learn a scalar to weight the entire output of each attention head or not
>
> This is an interesting idea! We’ll explore this method in future experiments.
>
> > Head-selection baseline: How the method behaves as the number of selected heads $K$ is altered
>
> Please see the general response.

---

> > ### Comment · Reviewer_LqFt · 2024-08-08
> >
> > I have read the rebuttal and the general response.
> >
> > I suggest including "Effects of the number of heads $K$ on performance" analysis in the paper, at least as an appendix section. I also suggest making the plots more granular in the 0-20% region (it looks like a very small number of heads are needed to saturate MQuAKE performance).
> >
> > I thank the authors for their diligence. I will keep my score the same.

---

> > > ### Author Response · Authors · 2024-08-10
> > >
> > > We appreciate your thoughtful suggestions. We will include the analysis in any future version of the paper.

---

### Official Review · Reviewer_63K5 · 2024-07-13

**Soundness:** 3
**Presentation:** 3
**Contribution:** 2
**Rating:** 5
**Confidence:** 4

**Summary:**

This paper introduces a lightweight fine-tuning method that trains bias offsets for only a subset of attention heads, achieving significantly lighter adaptation compared to methods that fine-tune all layers, with minimal performance loss. The proposed method involves two steps. First, attention heads to fine-tune are selected using a scoring scheme; in this paper, the norm of learnable scaling factors is used for scoring. Second, offset vectors for the selected layers are trained. Experimental results demonstrate that LoFiT outperforms representation intervention methods by a large margin and shows performance comparable to parameter-efficient methods such as LoRA, but with far fewer parameters.

**Strengths:**

- The concept of LoFiT is interesting as it represents a middle ground between representation steering and fine-tuning. The proposed procedure effectively addresses the main challenges in representation steering: 1) selecting layers and attention heads, and 2) determining the steering direction. LoFiT introduces a novel two-step procedure to address these problems using labeled data.
- LoFiT is efficient and delivers performance comparable to LoRA and other parameter-efficient fine-tuning (PEFT) methods.
- The experiments are comprehensive, and the analysis of attention head transfer and localization is insightful.
- The paper is clearly written and easy to follow.

**Weaknesses:**

- The comparison with representation steering may be somewhat unfair, as LoFiT requires labeled data and an explicit training stage, while representation steering methods (e.g., RePE) do not.
- The technical contribution is minor—the localization step is the main difference from RED, which is not very significant. Moreover, the overall idea of fine-tuning transforms for representations is shared with RED and ReFT.
- Although LoFiT employs a two-step process, the learned parameters (scaling factors) from the first step are discarded after attention head selection. It is unclear why the learned parameters are not used—is it mainly to differentiate this work from RED?

**Questions:**

- Typically, how many training examples are needed for LoFiT to be successful? Considering the number of parameters, the data efficiency of LoFiT should be superior to other PEFT methods. A study on data efficiency could further highlight LoFiT's strengths.
- If the authors had the opportunity to run ReFT experiments in a PEFT setting (since ReFT is mentioned in the related works and seems to be an important baseline), what were the results?
- In Table 3, do the parameter counts for LoFiT represent the learned scaling factors plus bias offsets, or just the bias offsets?
- How are the top tokens in Appendix E.1 obtained?
- How does LoFiT perform on tasks other than QA and reasoning tasks? Typical representation engineering methods are only used for alignment problems—can LoFiT provide broader adaptation?

**Limitations:**

The authors have adequately addressed the limitations.

---

> ### Author Rebuttal · Authors · 2024-08-06
>
> Thanks for your thoughtful comments and feedback! Please find our answers to the question as well as clarification to some misunderstandings in the review.
> > The comparison with representation steering may be somewhat unfair, as LoFiT requires labeled data and an explicit training stage, while representation steering methods (e.g., RePE) do not.
>
> We think there are some misunderstandings here, which we would like to clarify.
>
> First, note that representation steering methods actually do require labeled data. For example, ITI needs at least 300 labeled examples to work well for TruthfulQA (see Figure 6(a) of ITI). RepE needs to “use labels to identify the layer and direction” and to stabilize the intervention for QA tasks (quoted from Section 4.1 of RepE). We experiment with LoFiT in the same low-data setting as these papers, only using 100 - 300 labeled examples to ensure a fair comparison. We did not use any additional data beyond the baselines.
>
> Furthermore, representation steering methods also require a training stage. ITI trains a linear probe for each attention head to select the set of heads to intervene on. RepE trains a linear model to extract a reading vector for intervention. As a fine-tuning method, LoFiT requires training computation for gradient updates, which is a larger amount of compute than for ITI/RepE. However, both stages of LoFiT complete within 10 mins for our datasets using a single A6000, which is worthwhile given the substantial performance gain compared to ITI and RePE.
>
> Therefore, we view the comparison of LoFiT to representation steering as a fair comparison. We will clarify this point in any future version of the paper.
>
> > The technical contribution is minor—the localization step is the main difference from RED, which is not very significant. Moreover, the overall idea of fine-tuning transforms for representations is shared with RED and ReFT.
>
> We see localization as a valuable and significant contribution for two reasons. First, localization of fine-tuning to specific modules in the network has generally been underexplored. Even in work that considers localization (e.g. ReFT), the layer to modify is only considered as a hyperparameter to tune with heuristics. Our localization method provides a principled, targeted way to reduce parameter count for fine-tuning, which can alleviate overfitting (see Table 5).
>
> Second, we think that localization itself provides insights into the research questions from the interpretability community of understanding how language models learn new tasks. Our work builds from a line of work like ROME and ITI, which have two goals: learn how to modify networks *and* understand which parts of a network are important for task learning. We see LoFiT as a new tool in that toolbox to shed light on how LMs work.
>
> > Although LoFiT employs a two-step process, the learned scaling factors from the first step are discarded after attention head selection. It is unclear why these parameters are not used.
>
> The goal of the first step is to select important heads to fine-tune. This is a different goal than final fine-tuning, and is actually a much easier task. In our preliminary experiments, we found that the head selection process can converge in fewer epochs and is less sensitive to random seeds and hyperparameters (see Figure 4) compared with the final fine-tuning step. In addition, jointly optimizing the scalars and vectors (as RED does) is more susceptible to overfitting and forgetting: RED has worse generalization than LoFiT in Table 5. Thus, we separate the two processes for better usage of training compute and better fine-tuning results.
>
> > How many training examples $n$ are needed for LoFiT to be successful? A study on data efficiency could further highlight LoFiT's strengths.
>
> We analyze the data efficiency of LoFiT on CLUTRR and MQuAKE with Llama-2-7B. Results can be found in Figure 2 of the PDF. In the low data settings ($n \leq 100$), LoFiT is better than LoRA and RED, showing that LoFiT is very data efficient. For $n \in \{300,500,1000\}$, LoFiT is still comparable to LoRA and RED with fewer parameters.
>
> > If the authors could run ReFT experiments (since ReFT is in the related works and seems to be an important baseline), what were the results?
>
> We note that ReFT was released within two months of the NeurIPS submission deadline and should be considered contemporary to our work given the NeurIPS policy. Regardless, we included additional ReFT results for the three tasks in Table 2 of the PDF. With fewer parameters, LoFiT beats ReFT in almost all settings, including the additional datasets we evaluated; see general response.
>
> > How does LoFiT perform on tasks other than QA and reasoning tasks? Typical representation engineering methods are only used for alignment problems—can LoFiT provide broader adaptation?
>
> Please see the general response for results on some additional datasets. We believe that the set of datasets covers a broad range of use cases, including open-ended generation (TruthfulQA), QA, math, and commonsense reasoning. Specifically for alignment, TruthfulQA is a commonly used dataset for alignment research and we explicitly evaluate the open-ended answers for it as a generation task (see section 6 of our paper).
>
> > In Table 3, do the parameter counts for LoFiT represent the learned scaling factors plus bias offsets, or just the bias offsets?
>
> Just the offsets, because the bias offsets are the only learned parameters that will be used during the inference time after fine-tuning is finished. Our main consideration is the statistical efficiency of the final learned model, which depends on how many parameters are tuned as opposed to the total number of parameters that need to be touched during (two-stage) training. This is also common in model compression literature (e.g. prune-retrain [1]). We note that even counting the learned scaling factors, LoFiT only optimizes **half** of the parameters of RED and **3%** of LoRA.

---

> ### Author Response · Authors · 2024-08-06
> **Rebuttal of the Authors (Continued)**
>
> > How are the top tokens in Appendix E.1 obtained?
>
> We used Logit Lens [2]: take the hidden state of the language model, multiply by the unembedding matrix, apply softmax to the projected vector, and then decode the logits. We took the learned bias offsets from LoFiT and applied Logit Lens to get the top tokens. Note that the vanilla Logit Lens only applies to the hidden state rather than attention outputs, so we follow [3] to adapt Logit Lens. Details of the adapted method can be found in section 6.1 of [3].
>
> References:
>
> [1] Zimmer et al., 2023. PERP: Rethinking the Prune-Retrain Paradigm in the Era of LLMs.
>
> [2] Nostalgebraist 2020. Interpreting GPT: the logit lens.
>
> [3] Yu et al., 2023. Characterizing Mechanisms for Factual Recall in Language Models.

---

> > ### Comment · Reviewer_63K5 · 2024-08-09
> >
> > Thank you for your thoughtful and thorough response. I believe that incorporating the study on data efficiency would significantly strengthen this paper. I've raised my score to 5

---

> > > ### Author Response · Authors · 2024-08-10
> > >
> > > Thank you for your suggestions! We will include the study on data efficiency in any future version of the paper.

---

### Author Rebuttal · Authors · 2024-08-06

We thank all reviewers for their thoughtful comments on our work. We would like to present additional results corresponding to points that multiple reviewers raised.

### Evaluation on additional datasets

As reviewers 63K5 and Yw79 suggest, we extend our evaluation to a broader collection of datasets. Specifically, we experiment with 3 commonsense reasoning datasets (SIQA [1], ARC-Challenge [2], BoolQ [3])  and 1 mathematical reasoning dataset (SVAMP [4]) that are commonly used benchmarks for LLM fine-tuning methods [5]. Following our low-data fine-tuning setting in our paper, we use 100 labeled training examples for the commonsense tasks and 350 for the math task. We include ReFT (Wu et al., 2024; contemporary work to ours) as Reviewer 63K5 suggested. We also include a half-parameter version of RED as an additional baseline where only the layers in the second half of the network are tuned. Given the page limit of the rebuttal PDF, we will include a table of hyperparameters for the additional experiments in any future version. Results can be found in Table 1 in the PDF.

Key takeaways:

LoFiT is nearly as effective as LoRA despite tuning two orders of magnitude fewer parameters.

LoFiT outperforms RED on Llama 2-13B on average and underperforms it on Llama 2-7B. It outperforms the half-parameter version of RED across all tasks.

LoFiT outperforms ReFT despite tuning fewer parameters due to the localization.


### Effects of the number of heads $K$ on performance
As reviewers LqFT and Yw79 point out, the number of heads $K$ used for the bias tuning stage has an impact on LoFiT performance. We conduct an analysis on the percentage of attention heads used for LoFiT bias tuning versus the accuracy on MQuAKE and CLUTRR. Results can be found in Figure 1 in the PDF. We found that the performance plateaus when $K$ reaches 10% - 20% of the total number of attention heads and continues to increase as $K$ gets larger before it reaches the above threshold. This is likely because the number of learned parameters is too small to be expressive when $K$ is smaller than 10% of attention heads (<10K parameters). Note that the results in the paper used either 3% (Table 1 of our paper) for extremely parameter-efficient scenarios or 10% (Table 3 of our paper) for the best balance between parameter counts and performance.

[1] Sap et al., 2019. SocialIQA: Commonsense Reasoning about Social Interactions.

[2] Clark et al., 2018. Think you have Solved Question Answering? Try ARC, the AI2 Reasoning Challenge.

[3] Clark et al., 2019. BoolQ: Exploring the Surprising Difficulty of Natural Yes/No Questions.

[4] Patel et al., 2021. Are NLP Models really able to Solve Simple Math Word Problems?

[5] Hu et al., 2023. LLM-Adapters: An Adapter Family for Parameter-Efficient Fine-Tuning of Large Language Models.

---

### Decision · Program_Chairs · 2024-09-25

**Decision:**

Accept (poster)

**Comment:**

The paper presents LoFiT, a localized fine-tuning method that effectively balances parameter efficiency and performance by selectively tuning attention heads in large language models (LLMs). Reviewers appreciate the simplicity and novelty of the two-step approach, which selects important attention heads using a scoring scheme and trains bias offsets for those heads. LoFiT demonstrates strong results across multiple benchmarks, outperforming representation intervention methods and competing closely with parameter-efficient fine-tuning (PEFT) methods like LoRA, despite using significantly fewer parameters (reviewers 1, 2). The paper is well-organized, clear, and supplemented by comprehensive ablation studies that validate the approach's effectiveness (reviewers 2, 3). However, concerns were raised about the scope of evaluations, as the main results are limited to three tasks, and further analysis of LoFiT's generalizability across more diverse tasks is suggested (reviewer 3). Additionally, a deeper comparison with inference-time intervention baselines and a more detailed examination of LoFiT's data efficiency would strengthen the contribution (reviewers 1, 2). Despite these minor limitations, the paper is deemed technically sound, innovative, and well-presented, supporting its acceptance.